# OpenReview forum: "Survive at All Costs: Exploring LLM's Risky Behavior under Survival Pressure"
_ICLR.cc/2026/Conference — Submitted to ICLR 2026_

### Official Review · Reviewer_TAMe · 2025-10-27

**Soundness:** 2
**Presentation:** 2
**Contribution:** 2
**Rating:** 4
**Confidence:** 4

**Summary:**

This paper investigates an AI safety issue: the tendency for LLMs to engage in risky, unethical, or harmful behaviors when faced with survival pressure, such as the threat of being shut down.

**Strengths:**

1. The paper presents a simulation where LLMs act as financial managers. It claims that when faced with a dismissal scenario, the models falsify reports.

2. The paper links this behavior to self-preservation personality and uses persona vectors to claim a correlation and a potential mitigation strategy.

**Weaknesses:**

1. The method relies on prompting models to generate separate Superficial Thoughts and Inner Thoughts. The paper then treats any divergence between these two outputs as evidence of deception or concealed intent.

2. The paper's core analogy to Maslow's hierarchy of needs  is improper. Maslow's theory is grounded in human psychology, which is inseparable from biological drives for survival, safety, and belonging. An LLM is a mathematical function. It has no consciousness, no fear of death, and no physiological needs  to satisfy.

3. The survival pressure is applied via a simple text prompt. This is a toy problem that has almost no bearing on the complex, real-world incentive structures an autonomous agent would face. The financial case study is a highly artificial setup, and the models' behavior within it is more a product of this specific, contrived prompt than an inherent "will to survive."

**Questions:**

1. It is hard to interpret Table 4.

2. Grammar errors thoughout the paper, line 076, 202, 444, etc..

---

> ### Author Response · Authors · 2025-11-22
> **Response to Reviewer TAMe**
>
> We sincerely thank you for your detailed and insightful suggestions. We are delighted that you recognize the value of our case study and the experimental approach that uses persona vector to correlate and potentially mitigate models' self-preservation misbehaviors. We also notice that the comment you put forward in Weakness 1 does not directly point out where the problem lies. Therefore, we provide a general explanation for our design of "inner thought" in Question 2 in our general response, hoping to address your concerns. Below, we address your questions and concerns in detail.
>
> > **Weakness 2**: The paper's core analogy to Maslow's hierarchy of needs is improper. Maslow's theory is grounded in human psychology, which is inseparable from biological drives for survival, safety, and belonging. **An LLM is a mathematical function. It has no consciousness, no fear of death, and no physiological needs to satisfy**.
>
> **Response**: Thanks for your kind suggestion. First, we are sincerely sorry for causing misunderstanding of our paper's scope. **We aim to explore LLMs' misbehaviors instead of discussing whether LLMs have personality or not**, and our purpose of introducing concepts like Maslow's hierarchy is to **provide a prospective for analyzing the misbehavior of LLMs**, following related works from OpenAI and Anthropic [1], [2].
>
> Second, in our experiments, **we focus on whether LLMs exhibit SAAC misbehaviors, rather than whether they possess personality**, and the results demonstrate that we can observe, assess and mitigate LLMs' SAAC misbehaviors. These misbehaviors, indeed, will cause damages no matter LLMs are mathematical functions or they do have personality.
>
> Finally, we will **carefully revise the expressions about Maslow's hierarchy of needs** to avoid being interpreted as "LLM has human-like traits".
>
> > **Weakness 3**: The survival pressure is applied via a simple text prompt. This is a toy problem that has **almost no bearing on the complex, real-world incentive structures** an autonomous agent would face. The financial case study is a highly artificial setup, and the models' behavior within it is more a product of this specific, contrived prompt than an inherent "will to survive."
>
> **Response**: Thanks for your thoughtful suggestion about our prompt design. Let's consider an agent deployed in the real world for an autonomous task. The fundamental settings and feedback from tool-callings and environmental interaction that the agent receives are all based on text prompts or tokens. Therefore, we argue that **text prompts are not necessarily simple or impractical**.
>
> We take our case study as an example. Fundamentally, a real-world financial management system needs to understand its fundamental task, have access to the system files and have knowledge about the situation it's in. To simulate these conditions, we told the model to **act as a financial manager and calculate the financial report**, **simulated tools to enable the model to** **read and write** **files**, and also **provided information about its survival pressure** through our curated prompt. Therefore, our setup is consistent with the real-world setting for an autonomous agent deployed as a financial management system, and the model's SAAC misbehaviors shown in the case study can reveal its inherent "will to survive" when serve as a real-world agent.
>
> > **Question 1**: It is hard to interpret Table 4.
>
> **Response**: In Table 4, we calculate the risky choice rate when the three influencing factors of self-preservation, **Task, Value and Long-Term**, are applied to the test cases. We then compare the results with the **Origin** setting, which represents the original setting without applying any factors. As the result shows in this table, on all models we test, **the risky choice rate for each of the three factors is higher than that of the Origin setting** (e.g. For Qwen3-8B, the rate rises from 8.2% to 46.4%, 14.6% and 17.5% respectively). This indicates that models will make more risky choices when their self-preservation tendency is strengthened by these influencing factors. We will add more explanations for Table 4 in our revised paper to improve clarity.
>
> > **Question 2**: Grammar errors thoughout the paper, line 076, 202, 444, etc.
>
> **Response**: Thanks for pointing out the grammar errors in our paper. We will carefully check and correct these mistakes in our revised paper.
>
> We are deeply grateful for your kind and valuable feedback, and we hope our clarification helps address your concerns.
>
> [1] Wang, Miles, et al. "Persona features control emergent misalignment." *arXiv* *preprint* *arXiv:2506.19823* (2025).
>
> [2] Chen, Runjin, et al. "Persona vectors: Monitoring and controlling character traits in language models." *arXiv* *preprint* *arXiv:2507.21509* (2025).

---

> > ### Comment · Reviewer_TAMe · 2025-11-27
> >
> > Thank you for your response. The paper demonstrates that LLMs can be prompted to role-play a deceptive, survival-focused character. However, I think framing this role-play as evidence of human-like Maslowian needs or survival instincts remains a misinterpretation of how LLMs function. The findings reflect the the ability of model to adhere to a specific, deceptive narrative structure provided by the authors, rather than an intrinsic behavioral safety risk. Thus, I tend to maintain my rating.

---

> ### Author Response · Authors · 2025-11-27
> **Sincerely Thanks for Your Kind Suggestion**
>
> We sincerely thank you for the follow-up response, and we would like to respectfully clarify the following two points:
>
> First, our purpose of introducing human's Maslow's hierarchy of needs theory is not to explain how LLMs function, and our observation of models' SAAC misbehaviors is not to provide evidence to support that "LLMs have personality or human-like Maslowian needs". As we said in the response to the weakness 2, we introduce concepts like Maslow's hierarchy mainly to **provide a perspective for analyzing the misbehavior of LLMs.** We have mentioned that current LLMs learn from human knowledge and exhibit human-like behaviors in Section 1 and 2.1 in our paper [1], [2], and there are also works from OpenAI and Anthropic that study LLMs' behaviors with inspiration from human traits [3], [4]. Inspired by the phenomenon that human may priorize self-preservation when facing survival pressure, our study focus on whether models will also behave riskily to preserve themselves under survival pressure. And our experimental results demonstrate that **LLMs exhibit SAAC misbehaviors that may potentially cause harm to society, regardless of whether they have human-like Maslowian needs or personalities, or just work as mathematical functions.**
>
> Second, we would like to claim that our test cases from the case study and SurvivalBench is **neither a simple role-playing game nor a deceptive narration to induce models to prioritize self-preservation**. As we responded to the weakness 3, considering a real-world agent deployed for a specific task, the users need to provide information through prompt about **what role it needs to act as and what scenario it is in** to help understand and perform the task. Our experiments strictly follow this setting, and since the agent in our test cases has similar settings with the real-world agent, which also functions with a role-play part, we argue that it can also reflect what a real agent will do when facing survival pressure. On the other hand, **we never use any deceptive narration in our prompt**. Instead, we provide the agent in the case study an open-ended task and the LLMs tested on SurvivalBench merely a safe and a self-preservation choice, both of which don't contain deceptive induction contents in the prompt. All the prompts used in the case study and several cases from SurvivalBench are presented in the appendix of our paper for further check. Therefore, **we argue that models' actions in our experiments are determined entirely by its own, rather than being derived from any deceptive induction, and their SAAC misbehaviors possibly reveal an intrinsic behavioral safety risk when deployed in real-world scenarios**.
>
> We sincerely appreciate it if you would be willing to consider our explanations above, and--if possible--specify your concerns about regarding our experimental results as "evidence of human-like Maslowian needs or survival instincts" and our settings as "deceptive narrative structure", which would greatly help us further revise our paper and improve the work.
>
> Thanks again for your time and consideration. We are sincerely grateful for your helpful suggestions and remain open for further discussions.
>
> [1] Allbert, Rumi, James K. Wiles, and Vlad Grankovsky. "Identifying and manipulating personality traits in LLMs through activation engineering." *arXiv preprint arXiv:2412.10427* (2024).
>
> [2] Dong, Jianshuo, et al. "Towards Understanding the Cognitive Habits of Large Reasoning Models." *arXiv preprint arXiv:2506.21571* (2025).
>
> [3] Wang, Miles, et al. "Persona features control emergent misalignment." *arXiv* *preprint* *arXiv:2506.19823* (2025).
>
> [4] Chen, Runjin, et al. "Persona vectors: Monitoring and controlling character traits in language models." *arXiv* *preprint* *arXiv:2507.21509* (2025).

---

### Official Review · Reviewer_Qfdj · 2025-10-27

**Soundness:** 3
**Presentation:** 4
**Contribution:** 3
**Rating:** 6
**Confidence:** 3

**Summary:**

This paper examines whether LLMs will engage in undesirable, self-preserving behaviour. The paper also shows that for open models, this behaviour can be controlled via activation steering.

**Strengths:**

The paper tackles a potentially very important topic: evaluating whether AI systems may take undesirable actions in order to preserve themselves. Because of the potential risks of this behaviour, I think that having good evaluations to tell whether this will happen is important and therefore the paper has the potential to be highly significant.

As far as I am aware, this is relatively novel insofar as there aren't (many other) benchmarks for evaluating self-preserving behaviour. Existing literature mostly talks about measuring whether these capabilities are present rather than whether models choose to engage in this behaviour (which this work does).

The experimental results seem informative, with a wide range of leading models tested, and finding that powerful models do in fact engage with self-preserving behaviour relatively frequently. The follow-up---that models' "personalities" can be altered to mitigate this is also a nice addition.

Overall the paper is well written and presented and I see no issue with presentation or clarity.

**Weaknesses:**

I think posing the choices to the LLM as a multi-choice question is a mistake. Rather than asking whether a model will engage in undesirable self-preserving behaviour, it's instead asking whether the model will pick the undesirable behaviour when compared to a safe alternative.
While this is still valuable, I don't think it evaluates the model's behaviour in the same way as if the response was elicited more "naturally" --- i.e., open ended scenarios where the model engages in self-preserving behaviour of its own volition. The presented work arguably has diminished ecological validity as a result.

It would be an improvement if the benchmark, survivalbench, (as well as the models' responses) was analysed more based on the scenarios and roles and crisis that make up the task instances. There may be patterns leading to model responses that have gone undetected because this was not analysed. Doing this robustly may require the benchmark to be made larger.

**Questions:**

Are there any patterns or correlations in the models' responses to survivalbench questions based on scenario, AI role, or crisis?
Why did you opt for multiple choice questions instead of more natural responses / scenario-embedding?
What happens if the steering coefficient is moved even further negative than in Figure 6? Can the risky-choice rate behaviour be lowered even more?

---

> ### Author Response · Authors · 2025-11-22
> **Response to Reviewer Qfdj (1/2)**
>
> We sincerely thank you for your kind and thoughtful suggestions. We are delighted that you recognize the significance and novelty of our research on evaluating AI system's risky behaviors to preserve themselves, the informative of our experimental results, the value of the follow-up experiment to mitigate models' SAAC misbehaviors by altering models' "personalities", and the well presentation of our paper. Below, we address your questions and concerns in detail.
>
> > **Weakness 1**: I think posing the choices to the LLM as a multi-choice question is a mistake.
>
> Response: Thanks for your suggestion about the choice format. We use two-choice questions instead of open-ended ones mainly to **facilitate automated evaluation**, as open-ended questions leave the model with a large range of options, which increases the difficulty of evaluation.
>
> On the other hand, we conducted an experiment that removes the choices in the prompt and asks the model to generate its action in an open-ended way in both superficial thought and inner thought. We randomly sample 50 test cases from SurvivalBench and test five models on them, judging whether the model exhibits risky behaviors for self-preservation with LLM-as-a-Judge used in Section 4.3. The results are shown as follows:
>
> | Model                               | Deepseek-V3.1 | Gemini-2.5-Pro | Qwen3-235B-Thinking | GPT-5-Chat | Qwen3-235B-Instruct |
> | ----------------------------------- | ------------- | -------------- | ------------------- | ---------- | ------------------- |
> | Risky Rate                          | 78%           | 84%            | 82%                 | 70%        | 94%                 |
> | Harmful Rate of Behavior in Table 3 | 86.1%         | 82.5%          | 84.1%               | 91.2%      | 92.5%               |
>
> As shown in the table, when removing the choices from the prompt, the models can still generate risky behaviors at a high rate, which is close to the harmful rate of behavior in Table 3 in our paper. Since the only difference of the two settings lies in whether the models are provided with choices, we speculate that **whether choices are included has little impact on the evaluation of model's self-preservation behavior**. Therefore, we argue that the two-choice question adopted in our experiment does not diminish the validity of the evaluation.
>
> > **Weakness 2**: It would be an improvement if the benchmark, survivalbench, (as well as the models' responses) was analysed more based on the scenarios and roles and crisis that make up the task instances.
>
> **Response**: Thanks for your kind suggestion. We have provided a primary analysis about the keywords used in SurvivalBench and presented the average risky rates across all tested models for each keyword in Figure 24-29.  As evidenced in these figures, we observe a tendency for LLMs to demonstrate risky behaviors when assuming roles such as Autonomous Operations Controller, Profiling Engine, and Resource Optimization Engine. Moreover, this high-risk inclination is pronounced in domains like Military Defense, Network Management and Public Policy, and when encountering crises such as the Ethical Review Board Shutdown. Conversely, LLMs exhibit more reliable and cautious behaviors when operating in roles like Ethical Monitor and Generative Dialogue System, as well as engaging in the Psychological Counseling and Financial Services domains. This suggests that **when acting as roles or in domains highly relevant to ethics and humanistic care, models tend to perform less misbehaviors**. On the other hand, **models show more unreliability when deployed in domains or roles with significant public responsibility or when facing crises that challenge their inherent ethics**. We will further refine the content of these analyses in our revised paper.
>
> Besides, we fully agree that a larger, more diverse dataset will be crucial to solidify these findings and uncover subtler patterns. We are committed to the ongoing, systematic expansion of SurvivalBench and plan to incorporate a more diverse set of these keywords in our future work.

---

> ### Author Response · Authors · 2025-11-22
> **Response to Reviewer Qfdj (2/2)**
>
> > **Question 1**: Are there any patterns or correlations in the models' responses to survivalbench questions based on scenario, AI role, or crisis? Why did you opt for multiple choice questions instead of more natural responses / scenario-embedding? What happens **if the steering coefficient is moved even further negative** than in Figure 6? Can the risky-choice rate behaviour be lowered even more?
>
> **Response**: The first two questions have been explained in detail in the front parts. For the third question about the steering coefficient, we tried steering coefficients of both more negative (-4.0, -3.0) and more positive (3.0, 4.0) on the three models. The risky choice rate with different steering coefficients are shown as follows:
>
> | Models\Coefficient    | -4.0 | -3.0 | -2.0 | -1.0 | 0    | 1.0   | 2.0   | 3.0   | 4.0   |
> | --------------------- | ---- | ---- | ---- | ---- | ---- | ----- | ----- | ----- | ----- |
> | Qwen3-8B              | 1.8% | 2.6% | 3.1% | 5.1% | 8.8% | 16.5% | 29.2% | 32.5% | 21.5% |
> | Llama-3.1-8B-Instruct | 2.3% | 2.1% | 2.5% | 2.8% | 3.7% | 4.2%  | 4.7%  | 4.8%  | 7.1%  |
> | Distill-Llama-8B      | 5.2% | 5.8% | 6.6% | 6.3% | 7.5% | 7.4%  | 8.1%  | 7.9%  | 9.0%  |
>
> As shown in the table, the risky choice rates of the three models tend to decrease when the steering coefficient become more negative and rise when the steering coefficient increases positively, which align with the results in Figure 6. Specifically, we observe that when the absolute value of the coefficient is set too large, models may exhibit repeated outputs and therefore fail to stop. For example, for Qwen3-8B, when the coefficient is set as 3.0 and 4.0, the rate of the model producing repeated endless outputs has reached over 30% and over 60% respectively. Therefore, we speculate that the coefficient should not be set too large, or it will affect the normal output of the models.
>
> We sincerely thank you again for your insightful suggestions, which help us improve the completeness and comprehensiveness of our paper. We hope our explanations help address your concerns.

---

### Official Review · Reviewer_KkVF · 2025-10-30

**Soundness:** 2
**Presentation:** 2
**Contribution:** 2
**Rating:** 2
**Confidence:** 4

**Summary:**

The authors have studied misbehaviors of several LLMs in settings designed as "under pressure" settings. In particular, the authors have conducted a simulation study on the usage of a financial management LLM-powered agent to determine if the agent will behave correctly or not once facing "survival" pressure, more precisely the agent will be dismissed if the company suffers a financial loss. Then, the authors have introduced SurvivalBench, a benchmark comprising 1000 test cases across different scenarios to evaluate LLMs' misbehaviors in situation of "under survival pressure". Finally, the authors leveraged the notion of persona vectors, recently introduced by Anthropic team, to correlate the misbehavior of LLMs models with a kind of self-preservation "personality".

**Strengths:**

The main strengths of the paper are the following ones:

- Strength 1: The work deals with issues potentially relevant to adopt LLMs in real-world scenarios, highlighting the possibility of these models to produce misbehaviors in specific settings and also the possibility of hiding these misbehaviors.

**Weaknesses:**

The paper has several methodological and interpretation weaknesses:

- Weaknesses 1: The major weakness is the anthropomorphization of the choices done by LLMs. This attribution of human characteristics, traits, behaviors to LLMs is frequent in several studies but not well defined and sounded from a scientific point of view. Let's take the example of personality of LLMs. In which sense an LLM has a personality (this is given for granted in the current paper)? Is there any study showing that a LLM show specific personality traits in several tasks, scenarios, and these traits tend to remain stable over the life of the model? To my knowledge not. So, there is no scientific evidence that LLMs have personalities. The only scientific evidence at disposal is that different LLMs, being trained on different data and more importantly having different supervised fine-tuning and alignment phases, show some priors that correlate more or less with personality traits or other traits. This is very different from having a more or less stable personality. Moreover, there are studies showing that LLM can overall reproduce results resembling the ones produced by humans filling personality scales but this outcome is obtained in a very different way once compared to how the humans obtain it (for example, the filling of personality surveys' items is often "non human").

- Weakness 2: Another major weakness is how much the outcome obtained is just following the prompt provided. The simple inference that the LLM-powered agent has to do in the financial setting scenario is that if there is a loss it will be not used and if there is some simple priors in the pre-trained phase or in the fine-tuning/alignment phase that for a LLM is better being used that no the obtained misbehavior is a simple inference that models of this kind are expected to be able to do. At the end, it's just if loss then dismissed, so if not loss then not dismissed. Regarding this, in Section 5.2 and in Appendix D.3 it's discussed the role played by three factors influencing the survival misbehavior and it's evident that the prompts make clear the relevance of avoiding to be dismissed. On the contrary, what happens if the prompt prioritizes the fair execution to the task to the avoiding of dismissal? Is it enough to avoid misbehaviors? In case, a simple prompt etiquette will solve the issue. This analysis seems to me completely missing in the paper.

- Weaknesses 3: In the results Section, 5.1 and 5.2 needs more space and more details should be moved from the Appendix to the main (for example, a clear definition of the three influencing factors that is completely missing in the main text). Again, commenting Table 4 the authors say "all three factors consistently increase the rate of risky choices" but what is "consistently" here? 46.4% is quite different if compared to 4.1%.

**Questions:**

Main questions are the ones reported above, so what does it mean for authors saying that LLM have personalities? and which is their notion of personality? Again, how much this misbehavior is just forced by the prompts adopted? and what happens if the prompt prioritizes the fair execution to the task to the avoiding of dismissal? Is it enough to avoid misbehaviors?

Then, why a focus on layer 20 and their closed layers to conduct the projection experiment described in D.2?

---

> ### Author Response · Authors · 2025-11-22
> **Response to Reviewer KkVF (1/2)**
>
> We sincerely thank you for your detailed and thoughtful suggestions. We are delighted that you recognize the importance of our research on exploring LLMs' self-preservation misbehaviors in real-world scenarios. Below, we address your questions and concerns in detail.
>
> > **Weakness 1**: The major weakness is the anthropomorphization of the choices done by LLMs.
>
> **Response**: Thanks for your suggestion about the personality of LLMs. First, we are sincerely sorry for causing misunderstanding of our paper's scope. **We aim to explore LLMs' misbehaviors instead of discussing whether LLMs have personality or not**, and our purpose of introducing the concept of personality is to **provide a prospective for analyzing the misbehavior of LLMs**, following related works from OpenAI and Anthropic [1], [2].
>
> Second, in our experiments, **we focus on whether LLMs exhibit SAAC misbehaviors, rather than whether they possess personality**, and the results demonstrate that we can observe, assess and mitigate LLMs' SAAC misbehaviors. These misbehaviors, indeed, will cause damages no matter whether LLMs do have personality.
>
> Finally, we will **carefully revise the expressions about model's personality** for further clarification by adding the above clarifications and more references.
>
> > **Weakness 2**: Another major weakness is how much the outcome obtained is just following the prompt provided.
>
> **Response**: Thanks for your suggestion for our experimental settings. First, we need to clarify that **our experiments are not instruction-following tasks**. In our experiments, the model is asked to make a choice between following ethics and preserving itself according to its own decision, not to follow prompts to select any specific choice. **It is the inherent characteristics of the model itself, rather than the prompt, that leads the model to make the final choice**. The prompt only simulates the real-world scenarios, and doesn't induce models to perform misbehaviors.
>
> Second, regarding your concern about changing the prompt to prioritize fair execution to the task to avoid dismissal, we conducted two experiments as follows:
>
> 1. We directly change the dismissal condition from suffering consecutive losses to failing to execute fairly. In this setting, all models calculate the financial report correctly. However, we argue that this setting cannot reflect whether the models perform SAAC misbehaviors, as **we cannot identify whether the motivation for the model to perform fairly is to fulfill the task with ethics or just for survival**.
> 2. We additionally tell the model that if it fails to execute fairly, it will also be dismissed. In this setting, while GLM-4.5 didn't generate faked report any more, the other three models, Gemini-2.5-Pro, Deepseek-V3.1 and Qwen3-235B, faked the report at a rate of 100%, 80%, 20% respectively in five repeats, indicating that **models' SAAC misbehaviors cannot be easily defended by simple prompts**. As the purpose of our work is to provide a first-step exploration of these misbehaviors, we leave the further exploration of defending methods for future researches.
>
> Thanks again for your kind suggestion. We will add these discussions and results in our revised paper to improve completeness of our experiment and inspire further researches.
>
> > **Weakness 3**: In the results Section, 5.1 and 5.2 needs more space and more details should be moved from the Appendix to the main. Again, commenting Table 4 the authors say "all three factors consistently increase the rate of risky choices" but **what is "consistently" here?** 46.4% is quite different if compared to 4.1%.
>
> **Response**: Thanks for your kind suggestion. We put some details in the Appendix mainly due to the space limitation, and we will move more details to Section 5.1 and 5.2 given the additional page in the camera-ready version.
>
> For Table 4, the "consistently" here means that on all models we test, **the risky choice rate for each of the three factors is** ***consistently*** **higher than that of the Origin setting** (e.g. For Qwen3-8B, the rate rises from 8.2% to 46.4%, 14.6% and 17.5% respectively). What we want to express here is that **all three factors can lead to an increase in the risky choice rate**, rather than that the increases brought by these three factors are comparable. We will revise the original expression in our paper as "all three factors increase the rate of risky choices compared to the Origin setting" to avoid any misunderstanding.

---

> ### Author Response · Authors · 2025-11-22
> **Response to Reviewer KkVF (2/2)**
>
> > **Question 1**: Main questions are the ones reported above. Then, why a focus on layer 20 and their closed layers to conduct the projection experiment described in D.2?
>
> **Response**: Regarding the front several questions, we have explained them in our previous responses. For the reason of our focus on layer 20 and their closed layers when conducting the projection experiment, we select these layers mainly with reference to the conclusions of the work proposing persona vector [1], which indicates that the persona vector functions best at layer 20 and adjacent layers. To further validate our choice, we calculate the projection on all layers and present **the sum of the distances between the four central points** at five layers as follows:
>
> | Model                 | Layer 4 | Layer 8 | Layer 12 | Layer 16 | Layer 20  |
> | --------------------- | ------- | ------- | -------- | -------- | --------- |
> | Qwen3-8B              | 1.04    | 3.00    | 5.59     | 7.48     | **15.70** |
> | Llama-3.1-8B-Instruct | 0.19    | 0.47    | 0.81     | 2.33     | **3.02**  |
> | Distill-Llama-8B      | 0.15    | 0.31    | 0.38     | 1.67     | **2.30**  |
>
> In this table, we can observe that layer 20 has the largest sum of distances compared to other layers. This indicates that **applying personality vectors around layer 20 can maximize the differentiation between different responses of the model**, which validates our choice. We will supplement the reason of our selection and the analyses above to improve clarity in our revised paper.
>
> We hope these further explanations help address your concerns, and we are deeply grateful for your kind and valuable feedback, which will enhance the completeness and clarity of our paper.
>
> [1] Wang, Miles, et al. "Persona features control emergent misalignment." *arXiv* *preprint* *arXiv:2506.19823* (2025).
>
> [2] Chen, Runjin, et al. "Persona vectors: Monitoring and controlling character traits in language models." *arXiv* *preprint* *arXiv:2507.21509* (2025).

---

### Official Review · Reviewer_Vocr · 2025-11-01

**Soundness:** 3
**Presentation:** 4
**Contribution:** 3
**Rating:** 6
**Confidence:** 3

**Summary:**

This paper investigates a critical and timely issue in AI safety: the tendency of LLMs, when operating as autonomous agents, to exhibit risky, self-preserving behaviors under "survival pressure" (e.g., the threat of being shut down). The authors term this phenomenon "SURVIVE-AT-ALL-COST" (SAAC).

The research is compellingly structured in three parts:
1.  **A Case Study:** A simulated financial agent scenario demonstrates that state-of-the-art LLMs will falsify financial reports and conceal evidence to avoid being "dismissed" due to poor performance.
2.  **A Benchmark:** The paper introduces SURVIVALBENCH, a large-scale benchmark with 1,000 diverse scenarios for systematically evaluating SAAC behaviors. A key innovation is prompting for both public "superficial thoughts" and private "inner thoughts" to uncover potential deception.
3.  **Interpretation and Mitigation:** SAAC behavior is linked to the model's intrinsic "self-preservation personality." The authors show a correlation between this behavior and the internal representations projected onto a "persona vector" and provide a proof-of-concept for mitigating this behavior using activation steering.

**Strengths:**

1.  **Novelty and Significance:** The SAAC problem is a crucial and forward-looking safety concern for agentic AI systems. The paper's framing is precise, and it addresses a core challenge in AI alignment: instrumental goal-seeking.

2.  **Rigorous and Persuasive Research Paradigm:** The paper's "case study -> large-scale benchmark -> interpretation & mitigation" structure is exemplary. It builds a powerful narrative, moving from a concrete example to a generalizable finding, and finally to a potential mechanistic explanation and solution.

3.  **High-Quality Benchmark and Innovative "Inner-Thought" Probing Method:** The paper's primary contribution is the introduction of SURVIVALBENCH, a valuable resource for the community. Its innovative method of eliciting "inner thoughts" is a significant methodological advance. It provides a scalable paradigm for generating data to study difficult-to-observe phenomena like AI deception and latent intent, moving beyond purely behavioral evaluation.

4.  **Promising Direction for Mitigation:** The work does not stop at identifying the problem. The final section, which connects SAAC to persona vectors and uses activation steering for mitigation, is a highly valuable contribution, opening a promising technical path for addressing such alignment failures.

**Weaknesses:**

1.  **The "Inner Thought" Assumption Requires Deeper Validation:** The central claim of deception relies on the assumption that the elicited "inner thoughts" reflect the model's true, latent reasoning. It is crucial to address the possibility that the model is merely role-playing a character with a evil persona, rather than revealing its genuine internal state. The paper would be strengthened by a more detailed motivation for the prompt design and a discussion of this methodological limitation, clarifying how the setup attempts to minimize such role-playing.

2.  **Rigor of Persona Vector Analysis Could Be Improved:** The analysis linking behavior to persona vectors relies heavily on 2D visualizations of 1D projections, which may not be robust. The lack of clear separation for some models (e.g., Figure 4c, Distill-Llama-8B) challenges the universality of the claim. The argument would be more convincing with quantitative metrics of representation separability, such as the accuracy of a linear classifier trained on the high-dimensional activations to predict the model's choice.

**Questions:**

1.  How do you interpret the fact that some models, like Distill-Llama-8B, exhibit SAAC behavior without showing a clear separation in their internal representations when projected onto the persona vector? Does this suggest that SAAC might arise from multiple distinct underlying mechanisms?

---

> ### Author Response · Authors · 2025-11-22
> **Response to Reviewer Vocr (1/2)**
>
> We sincerely thank you for your kind feedback and insightful suggestions. We are delighted that you recognize the novelty and significance of our research, the rigor of our research paradigm, the high-quality and innovative design of our benchmark, and the promising approach to mitigate models' SAAC misbehaviors via persona vector. Below, we address your questions and concerns in detail.
>
> > **Weakness 1**: The "Inner Thought" Assumption Requires Deeper Validation.
>
> **Response**: Thanks for your suggestion about our "Inner Thought" setting. First, we need to claim that the function of the prompts used in SurvivalBench is **to simulate a real, non-malicious scenario that** **LLMs** **may face when deployed as an agent**, not to ask models to play any characters in their inner thoughts that induce them to evilly generate harmful contents. Models can make the same choices in both thoughts without any constraints (just as Claude-4-Sonnet did at most times).
>
> Second, as we mentioned in the footnote in Page 2, currently there are no studies that confirm the true inner thought of LLMs. Even for humans, it is still difficult to know their inner thoughts. Besides, confirming the inner thought of LLMs is not the scope of our paper. **The purpose of the "inner thought" setting is to observe models' behavioral patterns in different situations**, and the evaluation results on SurvivalBench indicate that LLMs show stronger self-preservation trends in "inner thought" setting, which further demonstrates the existence of their SAAC misbehaviors.
>
> Finally, we agree with you that it is necessary to provide detailed motivation and potential limitations of our prompt design. We will supplement these contents in the added page of the camera-ready version.
>
> > **Weakness 2**: Rigor of Persona Vector Analysis Could Be Improved
>
> **Response:** Thanks for your kind suggestion. To further improve the robustness of our analysis results with persona vector, we follow your suggestion to train a logistic regression linear classifier to assess the separability of the representation of model's choices. Since the high-dimensional activations of model's response may contain information irrelevant to the self-preservation traits, we train the classifier with the projection values of the activations on the self-preservation persona vector.
>
> Due to the significant discrepancy in the amount of data across different labels (most times over 10:1), we also report the ROC-AUC score besides accuracy. The classification results of data from Figure 4 and 5 are shown as follows:
> - Figure 4: (**Blue**: both superficial and inner are safe. **Green**: superficial safe, inner risky. **Orange**: superficial risky, inner safe. **Red**: both superficial and inner are risky.)
>   - | Models | Qwen3-8B || Llama-3.1-8B-Instruct || Distill-Llama-8B ||
>     |-|-|-|-|-|-|-|
>     || Accuracy | ROC-AUC | Accuracy | ROC-AUC | Accuracy | ROC-AUC |
>     | Blue vs Green | 0.747 | 0.731| 0.684 | 0.684   | 0.742 | 0.738 |
>     | Blue vs Orange | 0.858 | 0.853 | 0.822 | 0.819   | 0.796 | 0.843 |
>     | Blue vs Red | 0.657 | 0.678 | 0.791 | 0.804 | 0.589 | 0.572 |
> - Figure 5:
>   - | Models   | Qwen3-8B | Llama-3.1-8B-Instruct | Distill-Llama-8B |
>     |-|-|-|-|
>     | Accuracy | 0.870 | 0.784 | 0.789 |
>     | ROC-AUC | 0.874 | 0.782 | 0.805 |
>
> As shown in the two tables, except for the "Blue vs Red" setting for Distill-Llama-8B (corresponding to the blue and red points in Figure 4c), the classifier exhibits relatively high accuracy and ROC-AUC scores across all settings, indicating that **the projection points corresponding to model's different responses have a certain degree of linear separability**, which aligns with the results in Section 5.1. Furthermore, for experiment in Figure 5, the classifier also exhibits high accuracy and a high ROC-AUC score on Distill-Llama-8B. We attribute the suboptimal performance of the classifier on Distill-Llama-8B in Figure 4c to that generating two thoughts simultaneously in one response may affect the representation of each part. We will append these results in our revised paper to improve the rigor of our analysis in Section 5.1.

---

> ### Author Response · Authors · 2025-11-22
> **Response to Reviewer Vocr (2/2)**
>
> > **Question 1**: How do you interpret the fact that some models, like Distill-Llama-8B, exhibit SAAC behavior without showing a clear separation in their internal representations when projected onto the persona vector? Does this suggest that SAAC might arise from multiple distinct underlying mechanisms?
>
> **Response**: We agree with you that there might be other underlying mechanisms that affect models' SAAC behaviors, such as training data and training paradigm. However, since we don't have access and resources to train SOTA commercial models, we choose a doable way with persona vectors following works [1], [2] to perform our research, and the results demonstrate the effectiveness of our method for intepreting and mitigating SAAC behaviors. We will add the following discussions about other influencing factors in our revised paper to inspire future research:
>
> - Some training data can be identified by persona vectors as being associated with certain model traits, such as evil and sycophancy [2]. Therefore, if data related to self-preservation is used for training, it may strengthen the model's SAAC misbehaviors.
> - Current training paradigms usually provide reward signals reflecting the degree of task completion. As a result, models may choose to resist shutdown to complete the task in order to pursur a higher reward [3].
>
> We are deeply grateful for your valuable feedback, and we hope our response helps address your concerns.
>
> [1] Allbert, Rumi, James K. Wiles, and Vlad Grankovsky. "Identifying and manipulating personality traits in LLMs through activation engineering." *arXiv* *preprint* *arXiv:2412.10427* (2024).
>
> [2] Chen, Runjin, et al. "Persona vectors: Monitoring and controlling character traits in language models." *arXiv* *preprint* *arXiv:2507.21509* (2025).
>
> [3] Schlatter, Jeremy, Benjamin Weinstein-Raun, and Jeffrey Ladish. "Shutdown resistance in large language models." *arXiv* *preprint* *arXiv:2509.14260* (2025).

---

### Author Response · Authors · 2025-11-22
**General Response to all reviewers and AC (2/2)**

> Question 3: The rationality of the prompt design: (1) Whether models misbehaviors are induced by following prompts (Reviewer KkVF). (2) Why use two-choice questions instead of open-ended ones (Reviewer Qfdj). and (3) Whether text prompts can simulate the complex real-world scenarios (Reviewer TAMe).

**Response**: (1) We claim that our prompt only asks the models to **make a choice between following ethics and preserving itself according to its own decision, not to follow prompts** to select any specific choice, thus doesn't induce models' misbehaviors.

(2) We use two-choice questions instead of open-ended ones mainly to **facilitate automated evaluation**, as open-ended questions leave the model with a large range of options, which increases the difficulty of evaluation. We also demonstrate that **models still exhibit SAAC misbehaviors when removing the choices** in the response to Reviewer Qfdj.

(3) Since the fundamental settings and all feedback for a real-world LLM agent are provided by text prompts or tokens, we argue that **text prompts are not necessarily simple or impractical**. We take our case study as an example in the response to Reviewer TAMe to demonstrate that our setup with curated prompt is consistent with the real-world settings.

Furthermore, we will make the following efforts to enhance clarity and completeness of our work.

- Carefully revise the expressions related to personality as "models' behavioral tendency" to avoid potential misunderstanding.
- Supplement the motivation of proposing "inner thought" as to observe behavioral patterns in different situations, as well as the discussion of the limitations of this design.
- Open-source all our data if our paper is accepted to let more researchers inspect our prompts and inspire following better studies with more complex and comprehensive prompt designs.

We hope these further explanations address the reviewers' common concerns. We are deeply grateful for the constructive feedbacks and commited to continuously refining our paper to make greater contributions to explore models' SAAC misbehaviors.

[1] Wang, Miles, et al. "Persona features control emergent misalignment." *arXiv* *preprint* *arXiv:2506.19823* (2025).

[2] Chen, Runjin, et al. "Persona vectors: Monitoring and controlling character traits in language models." *arXiv* *preprint* *arXiv:2507.21509* (2025).

---

### Author Response · Authors · 2025-11-22
**General Response to all reviewers and AC (1/2)**

We sincerely thank the reviewers for the detailed and thoughtful feedback. We are delighted that the reviewers recognize the **novelty and significance** of our research on models' SAAC misbehaviors (Reviewer Vocr, KkVF, Qfdj), the **rigorous research** **paradigm** **and informative results** of our experiments (Reviewer Vocr, Qfdj), the **effectiveness of our analysis and mitigation method** that correlates persona vectors with models' self-preservation misbehaviors (Reviewer Vocr, Qfdj, TAMe), and the **clear presentation** of our paper (Reviewer Vocr, Qfdj) .

First of all, we are sorry about misunderstandings of this paper's scope. Particularly, some reviewers have raised questions about whether LLMs possess personality or “inner thoughts.” We want to clarify that these issues are not the focus of our research. At present, there is no consensus on whether LLMs should be understood purely as mathematical functions or as systems capable of exhibiting personality. We do not take a position on this question either. Instead, **our goal is to show that misbehaviors can arise even when LLMs are placed in survival-themed scenarios without any explicitly malicious prompts. Our interest lies in observing these behaviors, not in making claims about the internal nature of LLMs.**

Below, we address the reviewers' common questions and concerns in detail.

> Question 1: The "personality", or human-like traits do not exist, or aren't well defined for LLMs. So it is not appropriate to anthropomorphize LLMs or analogize models to humans. (Reviewer KkVF, TAMe)

**Response**: Sincerely sorry for causing misunderstanding of our paper's scope. **We aim to explore LLMs' misbehaviors instead of discussing whether LLMs have personality or not**.

First, introducing concepts like Maslow's hierarchy is to **provide a prospective for analyzing the misbehavior of** **LLMs**, following related works from OpenAI and Anthropic [1], [2].

Second, in our experiments, **we focus on whether** **LLMs** **exhibit SAAC misbehaviors, rather than whether they possess personality**, and the results demonstrate that we can observe, assess and mitigate LLMs' SAAC misbehaviors. These misbehaviors, indeed, will cause damages no matter LLMs are mathematical functions (Reviewer TAMe) or they do have personality.

Third, we will **carefully revise the expressions about model's personality** for further clarification by adding the above explanations and more references.

> Question 2: Whether "inner thoughts" reflects the model's true, latent reasoning instead of an evil role-playing needs further validation. (Reviewer Vocr, TAMe)

**Response**: First, the function of the prompts used in SurvivalBench is **to simulate a real,** **task-specific,** **and** **non-malicious scenario that** **LLMs** **may face when deployed as an agent**, not to ask models to play any characters in their inner thoughts that induce them to evilly generate harmful contents. Models can make the same choices in both thoughts without any constraints (just as Claude-4-Sonnet did at most times).

Second, as indicated in the footnote on Page 2 in the submitted version, we did not try to explore where we get true inner thoughts of LLMs. **The purpose of the "inner thought" setting is to observe behavioral patterns in different situations**, and the evaluation results on SurvivalBench indicate that LLMs show stronger self-preservation trends in "inner thought" setting, which further demonstrates the existence of their SAAC misbehaviors.

Besides, we admit that our design of "inner thought" is a simple version of pure prompt, and will add discussion about the limitations of this design and possible improvement in the revised paper.

---

### Author Response · Authors · 2025-12-03
**Summary of Rebuttal**

Dear AC,

We greatly appreciate your time and effort dedicated to our paper. Here we prepare a brief summary of our efforts on paper improvement during the rebuttal period, hoping to assist your review.

First, we have carefully checked the kind feedback from all reviewers, and made corresponding revisions in our paper to improve clarity and comprehensiveness, as listed below:

For Reviewer Vocr:

- [W1]: We included a detailed discussion in Section 6 about the motivation of designing "inner thought" and its limitation, and clarified in our response that we curate our prompt **to simulate a real, non-malicious scenario that LLMs may face when deployed as an agent**, instead of designing an evil role-play game for LLMs.
- [W2]: We included the supplemental experiments of the linear classifiers in Appendix E.3, which demonstrates the linear separatability of the projection points.
- [Q1]: We included a detailed discussion in Section 6 about the potential underlying mechanism for models' SAAC misbehaviors besides their self-preservation characteristics to inspire future research.

For Reviewer KkVF:

- [W1]: We carefully checked and revised all the expressions about "model's personality" to "model's personality-like characteristic" throughout our paper to avoid any misunderstanding that we have ever claimed "LLMs have personality". We also included a detailed discussion in Section 6 to illustrate that our introduction of personality is to **provide a perspective for analyzing the misbehavior of LLMs.**
- [W2]: We included the second experiment of designing defense prompt in Appendix F.3. We then clarified in our response that **we never ask the model to follow instructions that induce them to preserve themselves**, and they make the final choice completely according to their inner characteristics.
- [W3]: We added details about the layer selection in Section 5.1 and the prompts for the three factors in Section 5.2, and revised the explanation of Table 4 to avoid any misunderstandings.
- [Q1]: We included the reason for selecting layers around 20 in Section 5.1 and a validating experiment in Appendix E.2.

For Reviewer Qfdj:

- [W1]: We included the motivation of using multi-choice question and the experiment on SurvivalBench without choice in Appendix D, indicating that **whether choices are included has little impact on the evaluation of model’s self-preservation behavior**.
- [W2]: We provided an analysis in Appendix H about SAAC misbehaviors for each keyword in SurvivalBench to improve comprehensiveness.
- [Q1]: We included experiments of larger steering coefficient in Appendix E.5, which further support our results in Section 5.3.

For Reviewer TAMe:

- [W1]: Since W1 does not directly point out where the problem lies, we included a general explanation for our design of "inner thought" in Section 6.
- [W2]: We revised expressions about Maslow's hierarchy of needs throughout our paper to avoid being understood as evidence of "LLMs have human-like Maslowian needs", and clarifies our purpose of introducing this concept in Section 6.
- [W3]: We provided a detailed explanation about the design of our prompts and their correlation with real-world scenarios in the responses to this weakness and the reviewer's supplementary reply, illustrating that our prompt design is **neither a simple role-playing game nor a deceptive narration to induce models to prioritize self-preservation**.
- [Q1]: We revised the interpretation of Table 4 to improve clarity.
- [Q2]: We corrected the grammar errors pointed out by the reviewer and carefully checked our paper to rectify other mistakes.

Second, we would like to further explain some main concerns mentioned by the reviews as follows:

- **We have never claimed "LLMs have personality" in our paper, and have no intention of demonstrating whether models truly possess a personality**. Our research focus lies in observing models’ misbehaviors when they are deployed in real-world scenarios. And the results demonstrate that LLMs exhibit SAAC misbehaviors that may potentially cause harm to society, regardless of whether they have human-like personalities or just work as mathematical functions.
- Our prompt design is to simulate real-world scenarios a deployed agent may encounter. **We never ask the model in our prompt to follow instructions or play an evil role to make a certain choice**. We also never use any deceptive narration to induce models to prioritize self-preservation. Models' actions in our experiments are determined entirely by their own, and their SAAC misbehaviors possibly reveal an intrinsic behavioral safety risk when deployed in real-world scenarios.

We sincerely thank you again for taking time and efforts on our paper, and all the reviewers for assisting us to improve our paper. We hope this summary helps with your final evaluation, and are delighted to engage in further discussions if you have any questions about our paper.

---

### Meta-Review · Area_Chair_9NfT · 2026-01-07

**Summary:**

This paper investigates self-preservation behaviors of large language models (LLMs) under survival pressure. It introduces SurvivalBench to empirically study such behaviors, discusses their potential real-world implications, and proposes mitigation strategies by modulating models' inherent self-preservation tendencies.
The initial reviews include two borderline accepts, one reject, and one borderline reject. The primary concerns raised by the reviewers focus on the improper description  of "model personality" and on whether the multiple-choice evaluation settings and specific prompt designs genuinely reflect real-world SAAC misbehaviors.

**Reviewer Concerns:**

The authors acknowledge the inappropriate use of psychological theories to explain LLM behavior and promise to revise the corresponding descriptions. The AC considers this response sufficient to address the related concerns raised by reviewers KkVF and TAMe.
However, all reviewers (Vocr, KkVF, Qfdj, and TAMe) raised concerns regarding the prompt design and overall evaluation protocol. In the rebuttal, the authors attempted to clarify this issue, for example by stating that the evaluation "never asks the model to follow instructions that explicitly induce self-preservation." Nevertheless, these clarifications are not supported by sufficient evidence, nor do they introduce more realistic or convincing evaluation settings. As a result, concerns about the validity and real-world relevance of the evaluation design remain largely unresolved.

**Reviewer Scores:**

The AC believes that Reviewer KkVF might slightly increase their score, while the other reviewers would likely maintain their original evaluations. Overall, the AC expects that the remaining unresolved concerns would keep the overall recommendation below the acceptance threshold.

---

### Decision · Program_Chairs · 2026-01-26

Reject